# Impaired Awareness in People with Severe Alcohol-Related Cognitive Deficits Including Korskoff’s Syndrome: A Network Analysis

**DOI:** 10.3390/jcm12093139

**Published:** 2023-04-26

**Authors:** Hester Fidder, Ruth B. Veenhuizen, Ineke J. Gerridzen, Wessel N. van Wieringen, Martin Smalbrugge, Cees M. P. M. Hertogh, Anouk M. van Loon

**Affiliations:** 1Department of Medicine for Older People, Amsterdam Public Health Research Institute, Amsterdam University Medical Center, De Boelelaan 1109, 1081 HV Amsterdam, The Netherlands; i.gerridzen@atlant.nl (I.J.G.); m.smalbrugge@amsterdamumc.nl (M.S.); cmpm.hertogh@amsterdamumc.nl (C.M.P.M.H.); a.vanloon@amsterdamumc.nl (A.M.v.L.); 2Atlant, Nursing Home Markenhof, Korsakoff Centre of Expertise, Kuiltjesweg 1, 7361 TC Beekbergen, The Netherlands; 3Department of Epidemiology and Biostatistics, Amsterdam Public Health Research Institute, Amsterdam University Medical Center, De Boelelaan 1081a, 1081 HV Amsterdam, The Netherlands; w.vanwieringen@amsterdamumc.nl

**Keywords:** Korsakoff’s syndrome, nursing homes, network analysis, anosognosia, neuropsychiatric symptoms

## Abstract

Background: Impaired awareness of one’s own functioning is highly common in people with Korsakoff’s syndrome (KS). However, it is currently unclear how awareness relates to impairments in daily functioning and quality of life (QoL). Methods: We assessed how impaired awareness relates to cognitive, behavioral, physical, and social functioning and QoL by applying a network analysis. We used cross-sectional data from 215 patients with KS or other severe alcohol-related cognitive deficits living in Dutch long-term care facilities (LTCFs). Results: Apathy has the most central position in the network. Higher apathy scores relate positively to reduced cognition and to a greater decline in activities of daily living and negatively to social participation and the use of antipsychotic drugs. Impaired awareness is also a central node. It is positively related to a higher perceived QoL, reduced cognition and apathy, and negatively to social participation and length of stay in the LTCF. Mediated through apathy and social participation, impaired awareness is indirectly related to other neuropsychiatric symptoms. Conclusions: Impaired awareness is closely related to other domains of daily functioning and QoL of people with KS or other severe alcohol-related cognitive deficits living in LTCFs. Apathy plays a central role. Network analysis offers interesting insights to evaluate the interconnection of different symptoms and impairments in brain disorders such as KS.

## 1. Introduction

Korsakoff’s syndrome (KS) is a neuropsychiatric disorder that is caused by a deficiency of thiamine (vitamin B1), which is most often the result of alcohol abuse and malnutrition. No generally accepted definition of KS exists and different diagnostic classification systems are used [1]. It is thought to be the most common form of alcohol-related neurological damage and while worldwide studies on the prevalence of KS are scarce, the prevalence in the Netherlands has been estimated to be 4.8 per 10,000 inhabitants [2]. KS is characterized by impairments in different domains of functioning and amnesia is the main feature. Relative to other aspects of cognition, such as executive functioning, the impairment in memory is disproportionate [3]. However, severe alcohol-related cognitive deficits can be observed even in the absence of KS or other neurological complications [4]. Some argue that KS lies on a spectrum of alcohol-related cognitive disorders with overlapping diagnoses [5,6]. For patients with more widespread cognitive deficits, the term ‘alcohol-related brain damage’ is commonly used [4]. Neuropsychiatric symptoms (NPS) such as agitation, aggression, irritability, disinhibition and depressive disorders are highly prevalent in this group [7,8] and even symptoms of acute psychosis or catatonia may occur [9,10]. Besides NPS, somatic comorbidity is also typically present and physical functioning is often impaired [11]. Furthermore, KS affects the social domain: almost 75% of patients with severe KS experience loneliness [12] and have reduced social functioning [13]. Consequently, theirhe quality of life (QoL) is negatively affected [14]. Due to the complex nature of these problems and the specific care needs of patients, institutionalization in long-term care facilities (LTCFs) where multidisciplinary care is offered is often necessary.

Typically, people with KS have difficulties recognizing their own deficits, or “the patients believe that nothing is wrong with them”, as Egger stated [15]. To describe this state, multiple terms such as altered or impaired (self-)awareness, lack of insight, or anosognosia are used more or less interchangeably in the literature [16,17]. Recently, it was demonstrated that this phenomenon is highly common in people with KS living in Dutch LTCFs [18]. Furthermore, it has been described in various other neurological and neurodegenerative conditions, such as Huntington’s disease (HD), Parkinson’s disease (PD), stroke and traumatic brain injury (TBI) [19,20,21], but it has predominantly been studied in people with Alzheimer’s dementia (AD) [22,23,24]. In the context of dementia, Clare et al. use the term ‘awareness’, which they describe as “a reasonable or realistic perception or appraisal of a given aspect of one’s situation, functioning or performance, or of the resulting implications, expressed explicitly or implicitly” [25]. In this paper, we will adhere to this definition.

When awareness is impaired, it may influence a person’s functioning in different domains, which may have a significant impact on daily life, behavior and QoL [23]. People with impaired awareness may deny their behavioral change and may be limited in taking the perspective of another person. Furthermore, they may underestimate their cognitive impairment and overestimate their abilities. This may make them take part in activities beyond their true functional capacity and expose them to potential dangers like driving, leaving the gas on or refusal of care and institutionalization [26,27].

In people with KS or other forms of severe alcohol-related cognitive deficits, only the association between impaired awareness and NPS has been examined [18]. We hypothesize that in this group, impaired awareness is also associated with reduced functioning in the cognitive, physical, and social domains. Therefore, gaining an integral view of the complex interdependency of all these factors may be beneficial to a holistic understanding of the impact of this condition on people’s daily living. It might offer ways to provide more effective personalized care, for instance when refraining from arguing with those who are not aware of their own limitations and functioning.

Network analysis is a technique to visualize complex interactions between multiple variables that takes their mutual relationships and interdependence into account. It has emerged as an important technique in working fields like social sciences [28] and system biology [29]. It can help to understand the structure of a relationship, and it allows determining which variables play a central role by looking at how many connections to other variables exist. Furthermore, it is possible to evaluate how strongly variables are interconnected by looking at the strength of the relationships. A network analysis may help to concisely visualize complex datasets without the need for data reduction methods. It enables the researcher to represent interrelated statistical patterns in clear figures, showing substantial structures that are hard to extract by other means [30]. In psychopathology, network analysis is emerging as an alternative way of conceptualizing mental disorders. In this approach, mental disorders arise from interactions between symptoms that can be understood as a network [31,32,33]. Network analyses have also been applied to neurological disorders like Huntington’s Disease [34].

In this study, we applied a network analysis approach to explore the following research questions: I.How does impaired awareness relate to cognitive, behavioral, physical, and social functioning and to QoL in people with KS or other alcohol-related cognitive disorders living in Dutch LTCFs?  II.What other variables play a central role in the network?

We hypothesized that awareness might be connected to multiple other domains of functioning and might play a central role in the network.

## 2. Materials and Methods

### 2.1. Participants and Setting

We used an existing dataset from an observational, cross-sectional study by Gerridzen and colleagues [35]. These data were obtained between September 2014 and February 2016 from people with KS and other alcohol-related cognitive disorders living in 12 specialized LTCFs participating in the Dutch Korsakoff Knowledge Center (KKC) in the Netherlands. All residents had a primary diagnosis of KS or another alcohol-related cognitive disorder (Wernicke encephalopathy, Wernicke-Korsakoff’s syndrome, alcohol-related dementia, and alcohol-related brain damage) prior to admission to the LTCF. However, due to overlapping diagnostic profiles and diagnostic uncertainties between KS and other alcohol-related cognitive disorders, distinguishing these patient groups clearly is impossible.

The dataset contains data from 281 residents on a variety of topics such as sociodemographic and clinical characteristics as well as various measures (see: Section 2.2). Trained researchers and research assistants carried out the quantitative data collection, which consisted of a structured interview with the participants and/or professional caregiver (the primary responsible nurse or nurse assistant, or the attending elderly care physician) using questionnaires.

### 2.2. Measurements

To explore associations between awareness and functioning in the cognitive, behavioral, physical, and social domains and QoL, we used data from different questionnaires. All questionnaires used in the network analysis are described below. A more detailed description of all questionnaires is given by Gerridzen et al. [18]. Variables could only be included when they contained <15% missing data.

#### 2.2.1. Sociodemographic and Clinical Characteristics

Data on age, gender, education, marital status, length of stay (LoS) in the LTCF, and mainly used psychotropic drugs (antipsychotics, antidepressants, and benzodiazepines) was collected through a review of medical records.

#### 2.2.2. Awareness

Awareness of one’s own functioning was measured with the Patient Competency Rating Scale (PCRS) [36]. The PCRS is a 30-item questionnaire that covers the domains of activities of daily living, interpersonal functioning, everyday cognitive functioning, and emotional functioning. Each item is scored on a 5-point Likert scale (1 = “cannot do it,” 2 = “very difficult to do it,” 3 = “can do it with some difficulty,” 4 = “fairly easy to do it,” 5 = “can do it with ease”). Residents’ self-reported ratings (patient questionnaires) are compared to the professional caregivers’ questionnaires, which encompass the same questions. Total scores range from 30 to 150, and impaired self-awareness is deduced from discrepancies between these two ratings (discrepancy scores ranging from −120 to 120)., Positive discrepancy scores indicate overestimation and negative discrepancies indicate underestimation of functioning. Higher positive scores indicate higher levels of impaired awareness of someone’s functional deficits, with scores 0–28 indicating no or mildly impaired awareness, scores >28 indicating moderately impaired awareness, and scores >51 indicating severely impaired awareness).

#### 2.2.3. Neuropsychiatric Symptoms

NPS were assessed with the Neuropsychiatric Inventory Questionnaire (NPI-Q) [37], a 12-item questionnaire that assess the severity and associated caregiver distress of NPS. It is based on an interview with the first responsible nurse or nurse assistant. We categorized the following 8 NPI-Q items into five types of challenging behavior as defined by the Dutch Multidisciplinary guideline problem behavior in dementia [38], namely: anxiety symptoms (anxiety item), psychotic symptoms (delusions item/hallucinations item), depressive symptoms (depression/dysphoria item), agitation (agitation/aggression item/irritability item/disinhibition item), and apathy (apathy item). The severity scores were used. The total NPI-Q severity score represents the sum of individual symptom scores (range from 0 to 36 for 12 items; here, 0–24 for 8 items). Higher scores indicate more severe NPS. 

Since apathy was the only NPS that was previously associated with impaired awareness in KS [18], it was also individually measured with the nursing home (NH) version of the Apathy Evaluation Scale (AES-10) [39]. The AES is a 10-item questionnaire based on an interview with the professional caregiver. Answer categories range from 1 (= not at all characteristic) to 4 (= very characteristic), resulting in a scale ranging from 10 to 40. A higher total AES-10 score indicates more apathetic behavior. A score of 29 or less indicates the absence of apathy and a score of 30 or higher is indicative for apathy [40].

#### 2.2.4. Cognitive, Physical, and Social Functioning

The InterRai suite (RAI-LTCF) is a reliable multi-domain suite of assessment instruments developed for a wide range of vulnerable populations across settings, including LTCFs. It is based on an interview with the primary professional caregiver [41]. The InterRai contains scales to assess cognitive, physical, and social functioning.

#### 2.2.5. Cognitive Functioning

The Cognitive Performance Scale (CPS), a 6-item questionnaire, was used to measure cognitive functioning [42]. Scores range from 0 (=intact cognitive function) to 6 (= very severe cognitive impairment). Higher scores indicate lower cognitive functioning.

#### 2.2.6. Physical Functioning

Physical functioning was assessed with two scales: basic activities of daily living were measured with the Activities of Daily Living Self-Performance Hierarchy Scale (ADL), and instrumental activities of daily living were measured with the Instrumental Activities of Daily Living Performance and Capacities Scale (IADL) [43]. The ADL questionnaire includes questions about personal hygiene, locomotion, toilet use, and eating. Scores range from 0 to 6 and reflect the process of disablement by clustering ADL performance levels into stages of loss and higher scores thus indicate a greater decline (progressive loss) in ADL performance. 

As for IADL, the Interrai homecare (IRRS HC) version was used. Eight diverse tasks (meal preparation, housework, finances, managing medication, phone use, stairs, shopping, and transportation) were scored on both the performance and capacity scale. Scores range from 0 to 48, and higher scores indicate a higher level of functional independence.

#### 2.2.7. Social Functioning

Social functioning was measured using the Revised Index for social functioning (RISE), a 6-item questionnaire that assesses a person’s sense of initiative and social involvement in the facility. Scores range from 0 (poor social engagement) to 6 (high social engagement) [44].

#### 2.2.8. Quality of Life

QoL was assessed with the Manchester Short Assessment of Quality of Life (MANSA). This 12-item questionnaire explores satisfaction with life as a whole, using a 7-point rating scale (0 = extremely negative, 7 = extremely positive). Higher total scores indicate higher experienced QoL [45].

#### 2.2.9. Network and Statistical Analysis

To map and analyze the associations between all the variables, we conducted a network analysis. A network consists of nodes (variables) and edges (the associations between the variables). Furthermore, it is possible to evaluate how strongly nodes are interconnected by looking at the weight of edges (the strength). Variables can be connected either directly or indirectly (via other variables).

Out of all participants (N = 281), only participants with an available PCRS-score were included in the network analysis because awareness was the main focus of our study. Twenty other variables were included in the network (see above), visualized as nodes. The edges in the network analysis were derived from the data that were handled as described below. When missing data of a variable accounted for <15% of all observations, these variables were imputed using the MICE software package [46]. To normalize the data, we first transformed the data using a non-paranormal transformation with the HUGE software package. This transformation preserves the relationships among the variables and ensures distributional assumptions of the downstream methods. Next, we fitted a regularized Gaussian graphical model with a saturated underlying network to the transformed data using the R software package Rags2Ridges [47]. We chose the regularization parameter equal to 0.125, based on leave-one-out cross-validation (a method for resampling data that uses different samples of the data to test and train a model on different iterations). To remove spurious edges, we sparsified the saturated network by means of a false discovery rate (FDR) procedure [48] with a cut-off of 80%. To test the reliability of the resulting network, we re-ran the analyses (including imputation) 1000 times on a random subsample (75%) of the data and evaluated the selection frequency (i.e., occurrence in the sparsified network) and median partial correlation (i.e., association strength) of each edge. Finally, the network summary statistics eigenvector centrality, betweenness centrality, and degree centrality were calculated per node.

Eigenvector centrality measures a node’s importance while giving consideration to the importance of its neighbors. In all cases, the higher the score (1–0), the more central and important a node is considered to be in the network. Betweenness centrality indicates the number of times a node lies on the shortest path between other nodes. Degree centrality is defined as the number of edges connected to a node (number of relationships to other variables): The more connections, the more important a node is [30]. For the visualization of the network, we employed the Fruchterman–Reingold algorithm, which finds a graph layout such that the nodes are distributed to minimize the number of crossings between the lines connecting the nodes [49]. The thickness of the edges between variables in the network reflects the strength of the associations.

## 3. Results

### 3.1. Demographic and Clinical Characteristics

We included 215 patients with a completed PCRS. Table 1 shows demographic and clinical characteristics of these participants.

### 3.2. Network Analysis

We included 215 participants and 20 variables in the network analysis. Looking at the network (Figure 1), we can observe 26 edges (which hereafter will be referred to as associations or connections) between variables (Figure 1 and Table 2).

As can be seen in Figure 1 and Table 2, (AES) is the most interconnected variable (nodes) based on the number of connections and the centrality scores (a combination of Eigenvector, betweenness, and degree). Awareness (PCRS) is also a central node. Six variables, on the other hand, are only connected to one other variable, and the variable ‘single’ has no connection to any other variable.

As is graphically indicated by the thickness of the edges between variables in the network, some associations are stronger than others. The strength of the association indicated by the partial correlation and selection frequency of each association can be found in the Appendix A. The five strongest associations exist between (1) ADL and IADL, (2) cognitive impairment (CPS) and apathy (AES), (3) quality of Life (MANSA) and awareness (PCRS), (4) age and LoS, and (5) social participation (RISE) and apathy (AES).

As our aim was to explore relations between awareness (measured with the PCRS), other domains of functioning and QoL, we will focus on these associations below. Please note that the reported associations (positive and negative) are undirected, and no causality is indicated. We can merely infer that there is an association while taking into account all other connections in the network.

### 3.3. Associations with PCRS

The PCRS is directly connected to five other variables. As for the positive associations: A higher score on PCRS, indicating more impaired awareness of someone’s functional deficits, is directly positively associated with (1) a higher experienced quality of life (MANSA), (2) more impaired cognition (CPS), and (3) more of apathy (AES). In addition, we also observe negative associations between PCRS and (4) social participation (RISE), indicating that more impaired awareness is associated with less social participation, and (5) between PCRS and length of stay (LoS) in the LTCF, indicating that more impaired awareness is associated with a shorter period of living in the LTCF.

Finally, the five NPI-Q subscales are all indirectly related to PCRS, namely positively through AES and negatively through RISE, both via NPI-ap, which has the highest betweenness. The two scales that measure apathy, NPI-Ap and AES are positively associated. NPI-ap is directly related to NPI-agitation and NPI depression.

### 3.4. Associations with AES

While our primary research focus was on the role of awareness in relation to other domains, apathy (measured with the AES) also appears to play a central role in the network, as is reflected in the high centrality scores (highest Eigenvector and degree) (see Table 2). Therefore, we will discuss these associations with AES below.

Here, we observe a direct negative association between AES and RISE, indicating that more apathy is associated with less social participation. AES is also negatively connected to AP, indicating that more apathy is associated with the use of fewer antipsychotic drugs.

Moreover, we observe a direct positive association between AES and CPS, indicating that more apathy is associated with more impairments in cognition, as well as a direct positive association between AES and ADL and IADL. This indicates that more apathy is associated with a greater decline in functioning and higher dependence in (basic and instrumental) activities of daily living.

## 4. Discussion

In this study we explored the relations between impaired awareness of someone’s functional deficits and cognitive, behavioral, physical, and social functioning and QoL in patients with KS and other severe alcohol-related cognitive disorders living in Dutch LTCFs, using a network analysis. To our knowledge, we are the first to use this novel approach to gain insight into the influence of impaired awareness in different domains of functioning. In our network, impaired awareness was associated with multiple domains.

Impaired awareness was most strongly associated with QoL. This is in line with previous studies in patients with dementia and Huntington’s disease (HD). In people with dementia, anosognosia was associated with a better perceived QoL [50], especially in the advanced stages of the disease [51]. Likewise, in people with HD living in LTCFs, a positive association between impaired awareness and QoL was found, and the same questionnaires (PCRS and MANSA) were used as in our study [52].

Furthermore, in our network, impaired awareness was connected to impaired cognition and reduced social participation. Studies in people with AD have also shown that impaired awareness correlates to cognitive deficits and increases as the severity of cognitive impairment progresses [53,54]. In Parkinson’s disease, altered awareness of cognitive symptoms is common. People with cognitive impairments tend to under-report cognitive symptoms. No literature exists on the relationship between impaired awareness and social functioning in KS. However, Belfort and colleagues describe how caregiver-reported social and emotional functioning is associated with reduced awareness of social functioning in people with AD [55]. In general, social functioning and social cognition are known to be impaired in KS [13,56].

In some reports on AD, impaired awareness was found to increase over time, but in others, there was no relationship with variables such as the duration of illness [57,58]. We observed more impaired awareness with a shorter LoS in the LTCF. This might be explained in different ways. First of all, professional caregivers might have judged those patients that they know better (those who have been living in the LTCF for a longer period) differently, possibly assuming fewer impairments. The PCRS score is a discrepancy score, so this underestimation of functioning could have led to a lower discrepancy score, indicating less impaired awareness in the network. Another explanation that can be offered is that those patients who have been living longer in LTCFs have been confronted with their impairments more often and as a result have a more realistic view of their own functioning, as compared to those who have been living in the LTCF shortly or still live at home with no external feedback on their functioning.

### 4.1. Awareness and Apathy

Our finding that apathy and awareness are positively associated in the network is in line with previous studies in people with dementia. These studies showed that more impaired awareness was associated with more severe apathy and that the level of impaired awareness was found to be a significant predictor of apathy over time [57,59,60,61]. In KS, however, Egger et al. found that higher levels of insight (assessed through clinical judgment after team discussion) were associated with more apathy [15]. However, these results should be compared to ours with caution due to the small sample size and different methods used to assess and define awareness.

A possible explanation for the described association between apathy and impaired awareness might be that apathy causes a poor urge for introspection or self-examination, thus leading to poor self-insight (or impaired awareness), or vice versa, that poor self-insight does not often lead to self-examination and thus may trigger apathetic behavior. In addition, apathy and awareness have common frontal substrates that might explain the association [62]. More severe apathy may also have prevented people with KS from inclusion in this study, as non-completers of the PCRS patient’s form had more severe NPS [18]. Therefore, the association might have been stronger if these participants had been able to participate.

### 4.2. Apathy

Apathy was the most central node in our network. We observed that higher apathy scores relate positively to cognitive impairment and to more ADL dependence, and negatively to social participation and the use of antipsychotic drugs. These results are in line with a recent study that demonstrated that apathy scores correlated with overall cognitive function and were related to observer-rated daily executive problems [63]. Similarly, in people with AD, symptoms of apathy are associated with progression of cognitive symptoms [64], and Zhu and colleagues observed that in people with AD, apathy was independently associated with worse functioning [65].

Dorst and colleagues investigated the multidimensional nature of apathy in KS and concluded that social functioning is impaired, and that social apathy is most severe in KS (as compared to behavioral and emotional apathy) [66]. In line with this, KS patients have been described as ‘flat and emotionally detached’ [1] and Oey and colleagues describe that ‘emotional blunting’ (described as patients still experiencing emotions but being unable to display them) occurs in KS [67]. The negative association between apathy and social participation that we found might be explained by this.

Finally, we found a negative association between the use of antipsychotic drugs and apathy, indicating that more apathy is associated with the use of fewer antipsychotic drugs. A possible explanation for this association is that possibly fewer antipsychotic drugs were prescribed to residents who were already experiencing more apathy. Antipsychotic drugs are more often prescribed to residents who display other types of challenging behavior, such as agitation.

### 4.3. Strengths and Limitations

There are a few strengths and limitations to be addressed with regard to our study.

First of all, we are the first to apply network analysis to study impaired awareness in people with KS. Network analysis offers an explorative approach that is interesting because it enables us to evaluate the interconnection of different variables, taking into account the influence of all variables in the network. It can also help to generate new hypotheses for further research. This is especially interesting when studying impairments like awareness in brain diseases that have an impact on different domains, such as KS. Previous work by Borsboom and colleagues shows us that this approach can be suitable in psychopathology [31,32]. However, a limitation is that this network analysis is based on cross-sectional data and offers only an explorative approach to network structures and centrality parameters, but we cannot make deductions about causality from the resulting network.

Secondly, a strength of this study is the large sample size of people with a completed PCRS questionnaire (N = 215), especially when considering that data on this particular group (people with KS) are usually scarce. Unfortunately however, we had to deal with missing data. Out of the 281 patients that were included in the primary study, only 215 had completed the PCRS questionnaire (74%). Only the patients with a completed PCRS could be included in our network analysis. Those who had not completed the questionnaire did have higher NPS severity scores and demonstrated more difficulties in performing everyday tasks as judged by the professional caregiver with the PCRS [11]. This may have introduced a selection bias because this population might have scored higher on PCRS but also on NPI, CPS, (I)ADL and AES. As a result, these associations in the network might have been stronger, leading to different partial correlations and a different network.

As of yet, the dataset used in this analysis is the only one for the KS population in which any measure of awareness is reported alongside such a wide variety of other variables, as well as the only one of this size. However, we used the PCRS to assess impairments in awareness while multiple instruments to evaluate awareness exist and no consensus in the literature exists on which questionnaire is preferable [68,69]. Future research is recommended to validate the use of the PCRS in people with KS or to test whether other measures of awareness lead to comparable results. Finally, in this dataset, two different variables that represent apathy were used: the score of the AES questionnaire and the apathy subscale of the NPI-Q. However, the AES was used additionally, as it is a more valid, more extensive questionnaire on features of apathy than the NPI-Ap variable, which is only deduced from one item of the NPI-Q (“Does the patient seem less interested in his/her usual activities or in the activities and plans of others?”). Given that the NPI-Ap and AES did not correlate very strongly (see Appendix A), we believe that including these two questionnaires is valid since they did not measure exactly the same concept. Finally, because of the nosological uncertainties in the diagnosis of KS, our group of residents had diagnoses on the whole spectrum of alcohol-related cognitive disorders, varying from KS to alcohol-related brain damage and alcohol-related dementia, and we were not able to distinguish between these groups.

## 5. Conclusions

In this study we explored the relationships between impaired awareness and cognitive, behavioral, social, and physical functioning and QoL in patients with KS and other severe alcohol-related cognitive deficits living in Dutch LTCFs. Using a network analysis, we gained an integral view of the interconnections between these domains. As we hypothesized, impaired awareness is connected to many domains of functioning and QoL. Additionally, we found that apathy played a central role in these domains and that apathy is related to impaired awareness.

After this explorative network approach, more in-depth studies on how impaired awareness relates to these domains and its possible clinical implications are needed to gain a better understanding of this phenomenon. Further research could explore which clinical approaches are effective in dealing with impaired awareness in patients with KS living in LTCFs. When impaired awareness improves, this might have beneficial effects on other functional domains. However, it is arguable whether interventions to reduce impaired awareness are desirable, as people who are less aware of their situation tend to experience a better quality of life. For people with acute brain injury for instance, diverse intervention programs to improve awareness and maximize functional capacities such as neuropsychological programs have been developed in the past years. However, it is unclear what the long-term effects of these interventions are and whether these approaches would be suitable for our population [70].

In practice, patients with neurological diseases like KS and their (professional) caregivers may not always ‘be aware’ of the possible presence of impaired awareness and its negative consequences on different domains of functioning and QoL. We believe that being aware’ of the existence of impaired awareness in people with KS and other alcohol-related cognitive disorders might help caregivers to understand them better. Adapting care to the world of experience of people with KS by understanding altered awareness may offer opportunities to diminish neuropsychiatric symptoms and improve care.

## Figures and Tables

**Figure 1 jcm-12-03139-f001:**
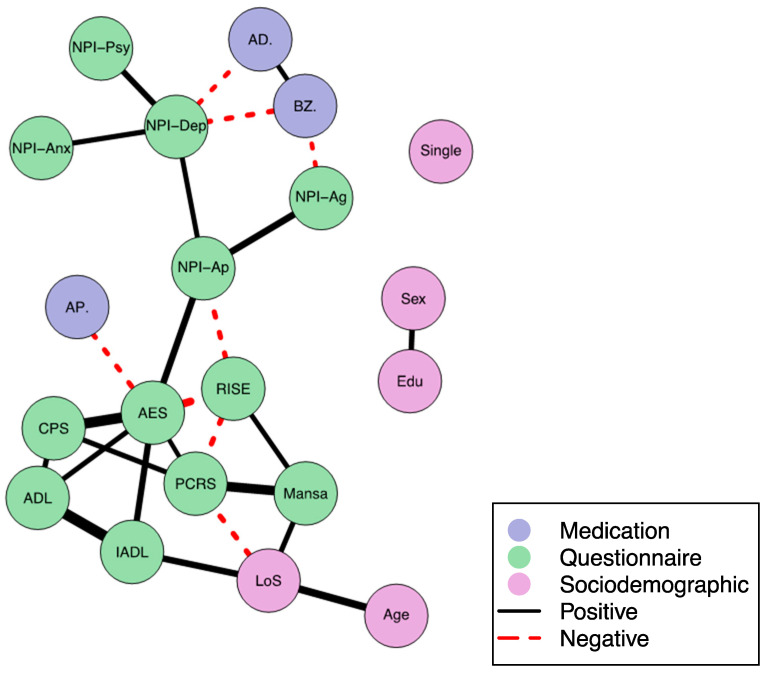
Network. Thickness of line indicates the median of the strength of the partial correlation. Abbreviations: AD. = use of antidepressants, ADL = activities of daily living, AES = apathy evaluation scale, AP. = use of antipsychotic drugs, BZ. = use of benzodiazepines, CPS = cognitive performance scale, Edu = level of education, IADL = instrumental activities of daily living, Los = length of stay in LCTF, Mansa = Manchester short assessment of quality of life, NPI = neuropsychiatric inventory, NPI-Ag = NPI agitation subscale, NPI-Ap =NPI apathy subscale, NPI-Anx = NPI anxiety subscale, NPI-Psy = NPI psychosis subscale, NPI-Dep = NPI depression subscale, PCRS = patient competency rating scale, RISE = revised index for social functioning.

**Table 1 jcm-12-03139-t001:** Demographic and clinical characteristics (N = 215).

Node Name	Variable	Mean [SD]	Count [%]	Missing [%]
	Sociodemographic and clinical characteristics			
Age	Age	63.2 [7.9]		0 [0]
Sex	Gender (male)		166 [77.2]	0 [0]
LoS	Length of stay in in LCTF (years)	6.7 [5.6]		0 [0]
Edu	Education (category)			31 [14.4]
	Elementary/lower		124 [67.4]	
	Secondary		43 [23.4]	
	Higher/university		17 [9.2]	
Single	Single, divorced, or widow(er)		188 [91.7]	10 [4.7]
	Use of ≥1 psychotropic drugs		137 [63.7]	0 [0]
AP.	Antipsychotic drugs		103 [47.9]	
AD.	Antidepressant drugs		79 [36.7]	
BZ.	Benzodiazepines		66 [30.7]	
	Questionnaires (↑: higher is better, ↓: lower is better)			
PCRS	Awareness discrepancy score (−120–120) ↓	40.0 [20.0]		0 [0]
AES	Apathy Evaluation Scale (10–40) ↓	25.3 [6.0]		2 [0.9]
NPI	Neuropsychiatric Symptoms (severity score) ↓			
NPI-Psy	Psychosis subscale			0 [0]
	Present		145 [67.4%]	
	Severity: 2 subscales (0–6)	0.86 [1.5]		
NPI-Ag	Agitation			0 [0]
	Present		175 [81.4%]	
	Severity: 3 subscales (0–9)	3.2 [2.6]		
NPI-Anx	Anxiety subscale			0 [0]
	Present		54 [25.1%]	
	Severity 1 subscale (0–3)	0.52 [1.0]		
NPI-Ap	Apathy subscale			0 [0]
	Present		98 [45.6%]	
	Severity: 1 subscale (0–3)	0.81 [1.0]		
NPI-Dep	Depression subscale			0 [0]
	Present		91 [42.3%]	
	Severity: 1 subscale (0–3)	0.82 [1.1]		
RISE	Social participation (0–6) ↑	4.27 [1.8]		0 [0]
CPS	Cognitive impairment (0–6) ↓	2.55 [1.6]		0 [0]
ADL	Activities of daily living (0–6) ↓	1.09 [1.2]		0 [0]
IADL	Instrumental activities of daily living (0–48) ↓	39.1 [8.3]		0 [0]
MANSA	Quality of life (12–84) ↑	61.0 [9.6]		3 [1.4]

**Table 2 jcm-12-03139-t002:** Network summary statistics: Centrality scores and number of connections.

	Centrality Scores			Number of Connections	
	Eigenvector	Betweenness	Degree	Positive	Negative
AES	1	74	7	5	2
CPS	0.679	0	3	3	0
ADL	0.622	0	3	3	0
IADL	0.567	0	3	3	0
PCRS	0.476	40	5	3	2
NPI-Ap	0.451	120	4	3	1
NPI-Ag	0.140	0	2	1	1
NPI-Dep	0.131	102	5	3	2
Mansa	0.099	32	3	3	0
LoS	0.074	32	4	3	1
NPI-Psy	0.032	0	1	1	0
NPI-Anx	0.029	0	1	1	0
Age	0.023	0	1	1	0
RISE	0	76	4	1	3
Sex	0	0	1	1	0
BZ.	0	8	3	1	2
AP.	0	0	1	0	1
AD.	0	0	2	1	1
Edu	0	0	1	1	0
Single	0	0	0	0	0

## Data Availability

The dataset used and analyzed during the study are available from the corresponding author on reasonable request.

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
