# Peer review of "Impaired Awareness in People with Severe Alcohol-Related Cognitive Deficits Including Korskoff’s Syndrome: A Network Analysis"

_jcm, 2023, doi:10.3390/jcm12093139_

Round 1

Reviewer 1 Report

This study aims at exploring how impaired awareness relates to cognitive, behavioral, physical and social functioning, as well as quality of life in a large dataset of patients with severe alcohol-related cognitive deficits (including KS) and using a network analysis approach.

The authors found that impaired awareness and apathy were the most central variables and that these variables were related with several clinical measures including notably perceived quality of life, cognition, social participation and length of stay in the nursing home.

The size of the dataset is very impressive, especially given the number of scales or questionnaires available. The novelty of the study relies also on the statistical approach, which is original but may need more explanation in the Results section (see below).

1)      My major problem concerns the systematic reference to KS, which is misleading since patients have severe alcohol-related cognitive deficits but not systematically KS. It is well recognized as a limitation. However,

·         the entire manuscript describes “KS patients”

·         the title refers to KS

·         the introduction refers to KS but without mentioning that amnesia is the most prominent and debilitating impairment in this disease, as well as the main feature

·         from the method section patients are referred to “KS patients”

The use of KS should be made more cautiously. For example, a suggestion of title: “Impaired awareness in patients with severe alcohol-related cognitive deficits including Korsakoff’s syndrome: a network analysis”. The introduction should mention severe alcohol-related cognitive deficits that can be observed even in absence of KS/neurological complication (ant not only KS).

Could you make subgroups? As previously mentioned, KS have amnesia at the forefront and do not recover. That specific profile could impact the results of the network analysis. For example, patients with executive dysfunction at the forefront may present a different level of awareness from those with severe episodic memory deficits at the forefront. It may be an error to generalize the findings of the present study to a specific clinical population i.e. KS.

2)                  Another suggestion to improve the manuscript is to favor the understanding of Table 2 (Lines 225-229). Given the numbers in the table, I understand why AES is central (eigenvector: 1, degree: 7) but not why PCRS is central (other variables have much higher eigenvector values and NPI-dep has as many connections). What is the “combination” (line 227) score that makes possible to say that awareness and apathy are the most central variables?

3)                  Lastly, Table 1 provides a lot of information regarding the characteristics of the dataset. Are there any cut-off scores for the scales and questionnaires that are used? That would help to interpret the different measures and to identify a general profile.

Minor concerns:

-The references are mainly Dutch oriented, which can make sense given the number of productive teams on the field of KS in the Netherlands. However, other classical references could be mentioned (just for examples, not mandatory):

https://pubmed.ncbi.nlm.nih.gov/35477346/

https://pubmed.ncbi.nlm.nih.gov/25307577/

-ADL in the abstract is not explained

-Line 34 “result of alcohol abuse and self-neglect”: why self-neglect? Do you refer to nutrition?

-Line 35: replace “brain disorders” by “neurological complication” since brain damage appears even in absence of KS

-Lines 37 to 39: state that memory deficits are disproportionate compared with other cognitive deficits in KS (see the classical description by Mickael Kopelman)

-Results are mainly discussed compared with the AD/HD literature. Are there data on other (non-degenerative) diseases?

-Line 306: specify the direction of the relationship between awareness and apathy in the present study.

-Lines 350-351, “it can help to generate new hypotheses for further research”: what are the new hypotheses based on the present results?

-I am not a native English speaker but I wonder whether the manuscript should be revised for English style (lines 281-284 for example) and to improve the punctation notably.

Reviewer 2 Report

* Comments and Suggestions for Authors (will be shown to authors)

1. The abstract could benefit from providing more details about the specific impairments in daily functioning and quality of life that the study examined. For instance, it could briefly summarize the key variables that were assessed and how they were measured.
2. The authors could consider providing more background information about Korsakoff syndrome and its prevalence, as well as how it relates to alcohol use disorder.
3. A) The introduction provides a strong foundation for the rest of the paper and sets the stage for the network analysis approach that was employed. While the paper does provide an estimate of the prevalence of Korsakoff syndrome in the Netherlands, it may be helpful to provide a global perspective on the prevalence of the disorder. This could help contextualize the findings and make them more applicable to a broader audience. B)  While the paper does provide a clear hypothesis regarding the interconnections between impaired awareness and other variables, it may be helpful to explicitly state the research questions that were addressed in the study. This could help readers better understand the purpose of the study and how the findings were obtained.
4. Methods:  it is not clear how missing data were handled for the variables used in the network analysis. The authors could provide more detail about any imputation or deletion methods used to handle missing data. Finally, it would be useful to include a flowchart of the study population, detailing the number of participants excluded and included at each stage of the study.
5. 2.4.2. Neuropsychiatric symptoms: The authors should consider citing a few cases or papers related to NPS, WE and Korsakoff syndrome to support the importance of measuring NPS in this population. Please consider major/ or common psychiatric complications/association  of WE/KS with depression, anxiety, psychosis, catatonia. 1) Wernicke encephalopathy in patients with depression: A systematic review Erik Oudman, PhD, 2) Wernicke’s Encephalopathy Mimicking an Acute Psychotic Disorder Li-Fen Chen, M.D., and Ching-En Lin, M.D. 3) An Unusual Presentation of Catatonia in Non-alcoholic Wernicke Encephalopathy Saeed Ahmed 1 2, Tayo V Akadiri, 4) A case of chronic Wernicke’s encephalopathy: a neuropsychological study imageErik Oudman
6. In fact line # 39,40,42, consider adding more references, common and rare neuropsychiatric complications ( as mentioned above in comment #5) and add more rare additional evidence for the validity and relevance of using the NPI-Q to assess NPS in this population.
7. 2.4.8. Network and statistical analysis: Great work, nicely done to map and analyze the associations between all the variables related to impaired awareness, daily functioning, and quality of life in people with Korsakoff syndrome living in specialized long-term care facilities. It might be useful for the authors to clarify why they chose the specific regularization parameter and FDR cut-off that they did. Additionally, it might be helpful for readers if the authors briefly explain what the Fruchterman-Reingold algorithm is and how it was used to visualize the network.
8. The discussion mentions the association between impaired awareness and QoL in KS, AD, and HD. Can authors explain why there might be a positive association between impaired awareness and QoL in advanced stages of dementia?
9. The discussion also mentions the possible explanations for the association between apathy and impaired awareness, such as common frontal substrates. Can authors provide more details about this and how it relates to KS specifically?
The authors did an amazing job! Their findings on the centrality of impaired awareness and apathy in the network, and their association with multiple domains of functioning, are particularly noteworthy. However, there's still some room for improvement in the study. The relatively small sample size and limited geographical scope of the study may limit how generalizable the findings are. Also, focusing only on patients living in long-term care facilities might not capture the full spectrum of the disorder, and future research could benefit from including patients living independently or with family members.

Round 2

Reviewer 2 Report

The authors have addressed all queries and concerns;